# INDOOR-3.6M: A MULTI-MODAL IMAGE DATASET FOR INDOOR GEOLOCATION

## ABSTRACT

Indoor image geolocation, the task of determining the location of an indoor scene based on visual content, presents unique challenges due to the constrained and repetitive nature of indoor spaces. Current geolocation methods, while advanced in outdoor contexts, struggle to perform accurately in indoor environments due to the lack of diverse and representative indoor datasets. To address this gap, we introduce INDOOR-3.6M, a large-scale dataset of geotagged indoor imagery spanning various residential, commercial, and public spaces from around the world. In addition to the dataset, we propose a new sampling methodology to ensure geographic diversity and balance. We also introduce INDOOR-15K, a benchmark for evaluating indoor-specific geolocation models. Finally, we demonstrate the dataset's utility by finetuning GeoCLIP using our dataset, which shows significant improvements over the GeoCLIP baseline on our test set and other benchmark test sets.

## 1 INTRODUCTION

Image geolocation, which involves determining the geographic origin of a photograph based on its visual content (Hays & Efros, 2008), is a critical vision task with a wide range of applications, including forensic investigations and fraud detection. Current approaches to geolocation typically follow either a retrieval-based or classification-based framework. Retrieval-based methods depend on extensive geotagged image databases, employing similarity metrics to match query images with known locations (Hays & Efros, 2008; Vo et al., 2017). In contrast, classification-based approaches discretize the Earth's surface into geocells, treating geolocation as a multi-class classification task, requiring substantial training data per geocell to achieve high accuracy (Seo et al., 2018; Weyand et al., 2016).

As with other core computer vision tasks—such as object detection, semantic segmentation, scene recognition and image classification—-the performance of geolocation models is closely tied to the availability of large, diverse, and high-quality datasets. Datasets such as ImageNet (Krizhevsky et al., 2017), MS COCO (Lin et al., 2014), and Places (Zhou et al., 2017) have been pivotal in driving progress in their respective domains. However, for image geolocation, the need for comprehensive datasets is even more pronounced due to the task's inherent complexity and global scope. The visual appearance of locations can vary dramatically depending on factors such as seasonal changes, time of day, weather conditions, and human-induced modificationsPramanick et al. (2022). Additionally, the global nature of the task requires representation across a wide variety of geographic regions, each

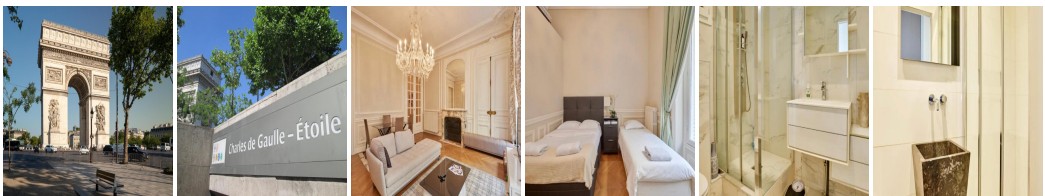

Figure 1: Images (a) and (b) show outdoor landmarks near the Arc de Triomphe, while (c)–(f) depict diverse indoor scenes from a nearby hotel. This highlights the geolocation challenge posed by visually similar indoor environments compared to distinctive outdoor environments.

possessing unique and sometimes subtle visual characteristics. Fine-grained geolocation further necessitates high-density, geotagged imagery to achieve precise localization.

To support the training and evaluation of geolocation models, high-quality geotagged images annotated with precise geographic location information are essential. The coverage, diversity, and geographic balance of these datasets directly influence the generalizability and accuracy of geolocation models in various contexts. Despite the growing availability of geotagged imagery from social media, curating datasets that are comprehensive, balanced, and representative of global geographic diversity remains challenging. Urban areas and popular tourist destinations are often over-represented, while rural or less-frequented regions suffer from data scarcity, leading to models that are less effective in underrepresented areas.

While significant advances have been made in outdoor and mixed-environment (or hybrid) image geolocation, indoor geolocation remains under-explored and presents a unique set of challenges. Unlike outdoor environments, where landmarks, street signs, skylines, and natural features offer rich contextual cues, indoor spaces are more constrained and visually repetitive. The interiors of buildings, rooms, and enclosed areas typically lack the expansive contextual markers found in outdoor settings. Moreover, variations in design, layout, and lighting across indoor spaces introduce additional layers of complexity. These factors highlight the need for datasets specifically tailored to indoor environments.

An indoor image typically depicts a scene from an enclosed or semi-enclosed space, such as a home, office, or public building, and is defined by elements like furniture, walls, artificial lighting, and interior structural elements. These spaces can range from small rooms to vast halls, each with distinct characteristics. The line between indoor and outdoor environments can also blur in transitional spaces like covered patios or parking garages, where structural openness is combined with indoor elements such as artificial lighting and furniture, producing environments that straddle the boundary between the enclosed and the open.

The feasibility of indoor image geolocation lies in the distinctive visual markers inherent in the design, utilization and layout of interior spaces. Regional, cultural, religious, economic, and political factors shape architectural styles, materials, decor, and spatial layouts, resulting in distinct visual characteristics that vary geographically. Furniture, decor, artwork, religious symbols, and fixtures like electrical outlets provide valuable locational clues. Additionally, the layout of indoor spaces is often tailored to human needs and influenced by local aesthetics, making their visual structure identifiable and learnable. Despite lacking the prominent landmarks typical of outdoor settings, indoor environments offer a rich array of details that can support effective geolocation.

Given the current emphasis on outdoor geolocation and the limited focus on indoor environments, it is evident that a geographically diverse dataset dedicated to indoor image geolocation is crucial for advancing this field. Such a dataset would capture the unique characteristics of indoor spaces across a broad range of geographic locations and functional areas. Its development represents a critical step toward addressing the existing gap in indoor geolocation research and enables the creation of models capable of fine-grained localization of complex, enclosed environments. To empower research into indoor image geolocation we make the following contributions:

- We introduce *INDOOR-3.6M*, a dataset of geotagged indoor imagery featuring diverse living spaces, functional areas, leisure and public facilities. This extensive collection, enriched with comprehensive multimodal metadata, will empower indoor-specific geolocation research, addressing a critical gap in the current literature.
- We propose a sampling strategy that offers a method for obtaining geographically representative samples from geographically biased datasets. Our approach considers both land area and population distribution, ensuring a balanced representation of images.
- We present INDOOR-15K, a geographically representative benchmark test set designed to evaluate the performance of both indoor-specific and hybrid geolocation models on diverse indoor scenes. This benchmark provides a standardized evaluation framework for fairly evaluating and comparing advancements in indoor geolocation research.
- Finally, we finetune GeoCLIP–yielding a specialized GeoCLIP model that outperforms the Geo-CLIP baseline across all levels of geographic granularity, establishing a benchmark for indoor image geolocation and paving the way for future innovations in this field.

The dataset along with evaluation scripts are available at: https://github.com/anonymous-for-double-blind-review.

## 2    RELATED WORK

In recent years, image geolocation has seen remarkable advancements, driven by a convergence of cutting-edge computer vision techniques, deep learning architectures, and the availability of large-scale geotagged image datasets. These innovations have significantly improved the ability of models to accurately predict the geographic origin of images. The evolution of geolocation techniques has largely been defined by two primary paradigms: retrieval-based approaches and classification-based approaches. While retrieval-based methods rely on matching query images with similar images in a large geotagged database, classification-based methods divide the Earth's surface into discrete regions or geocells (Weyand et al., 2016), treating geolocation as a multi-class classification problem. More recently, hybrid approaches have emerged, combining the strengths of both paradigms to enhance geolocation accuracy.

State-of-the-art systems like PIGEON/PIGEOTTO (Haas et al., 2023) and Geoclip (Vivanco Cepeda et al., 2024) exemplify this advancement. These models utilize CLIP Vision Transformers (ViTs) (Dosovitskiy et al., 2020; Radford et al., 2021) and leverage large-scale geotagged image datasets to infer geographic locations based on visual content. The success of these systems highlights the effectiveness of modern neural architectures in capturing complex visual features tied to specific geographic locations, and the importance of combining such architectures with comprehensive, high-quality datasets.

Table 1: Comparison of geolocation datasets. The "Benchmark" column indicates whether the dataset provides a dedicated test or evaluation set specifically designed to benchmark the performance of geolocation models.

| Dataset | Year | Size | Scene Type | Scale | Type | Bench-mark |
|---|---|---|---|---|---|---|
| Im2GPS (Hays & Efros, 2008) | 2008 | 6.5M | Mixed | Global | Geotagged | x |
| San Francisco Landmarks (Chen et al., 2011) | 2011 | 1.1M | Outdoor | City | Geotagged | x |
| YFCC100M (Thomee et al., 2016) | 2016 | 100M | Mixed | Global | Geotagged, Multimodal | x |
| MP-16 (Larson et al., 2017) | 2017 | 5M | Mixed | Global | Geotagged | x |
| PlaNet (Weyand et al., 2016) | 2016 | 126M | Outdoor | Global | Geotagged | x |
| Im2GPS3k (Vo et al., 2017) | 2017 | 3k | Mixed | Global | Geotagged | ✓ |
| YFCC4k (Vo et al., 2017) | 2017 | 4K | Mixed | Global | Geotagged | ✓ |
| YFCC26k (Muller-Budack et al., 2018) | 2018 | 26K | Mixed | Global | Geotagged | ✓ |
| Hotels50K (Stylianou et al., 2019) | 2019 | 1M | Indoor (Hotel rooms) | Global | Geotagged | x |
| GWS15K (Clark et al., 2023) | 2023 | 15K | Outdoor | Global | Geotagged | ✓ |
| **INDOOR-3.6M** | **2024** | **3.6M** | **Indoor (Scene agnostic)** | **Global** | **Geotagged, Multimodal** | x |
| **INDOOR-15K** | **2024** | **15K** | **Indoor (Scene agnostic)** | **Global** | **Geotagged, Multimodal** | ✓ |

Despite the remarkable advancements in image geolocation, global-scale geolocation remains a significant challenge, pushing researchers to focus on a more limited scope of the problem by directing attention towards closed-domain geolocation tasks. This shift arises due to the difficulties of tackling geolocation on a global scale, which necessitates access to an extensive, diverse, and truly global dataset-—an asset that remains elusive. As a result, researchers have concentrated on more

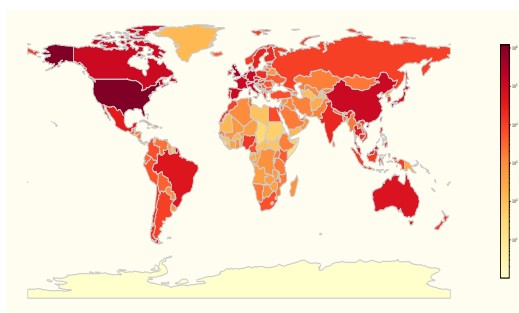 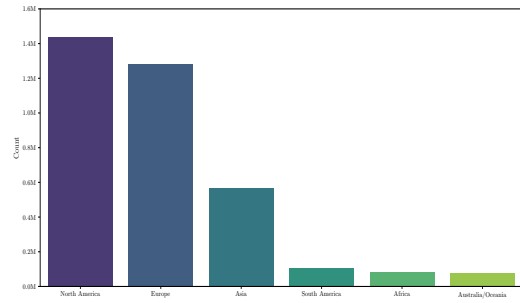

(a) Global distribution of INDOOR-3.6M data       (b) Distribution of Images Across Continents

constrained tasks such as geolocating images of skylines (Ramalingam et al., 2010), beaches (Cao et al., 2012), deserts (Tzeng et al., 2013), the Alps (Saurer et al., 2016), hotel rooms (Stylianou et al., 2019), or specific urban areas like San Francisco (Berton et al., 2022), or even individual countries like USA Suresh et al. (2018), by leveraging tailored datasets. While these focused efforts have yielded impressive results and enhanced our understanding of geolocation techniques, they leave an important gap in the field—-specifically, the geolocation of scene-agnostic indoor imagery on a global scale.

Indoor image geolocation presents a unique challenge with valuable applications in fields such as digital forensics, law enforcement, and augmented reality. However, it requires geotagged indoor imagery with global diversity, which current datasets lack. For instance, indoor datasets like NYU Depth V2 (Silberman et al., 2012), SUN RGB-D (Song et al., 2015), and Places365 (Zhou et al., 2017) are designed for tasks such as object detection and scene recognition but do not provide the geographic metadata necessary for geolocation. Similarly, mixed-environment datasets such as MediaEval Placing Task (MP-16) (Larson et al., 2017) and YFCC100M (Thomee et al., 2016), which encompass both indoor and outdoor environments, also fall short for indoor geolocation as they tend to prioritize outdoor scenes. While these datasets have led to the development of powerful geolocation algorithms, models trained on them often perform poorly on indoor imagery due to the substantial differences in visual characteristics between indoor and outdoor environments.

Existing image geolocation benchmark datasets, such as IM2GPS (Hays & Efros, 2008) and its successor IM2GPS3k (Vo et al., 2017), along with subsets of YFCC100M like YFCC4K (Vo et al., 2017) and YFCC26K (Muller-Budack et al., 2018), have been instrumental in evaluating geolocation systems. However, these datasets predominantly comprise outdoor imagery, rendering them inadequate for assessing indoor-specific geolocation tasks. Indoor environments present unique challenges, necessitating the interpretation of more complex and nuanced visual features, including variations in room layout, furniture arrangements, lighting conditions, and decorative elements. Consequently, to facilitate accurate indoor geolocation, it is imperative to develop specialized indoor-specific datasets for both training and benchmarking purposes.

## 3 DATASET OVERVIEW

To achieve accurate and reliable indoor image geolocation, a large and diverse dataset covering various indoor environments is essential. The INDOOR-3.6M dataset addresses this need by being agnostic to specific indoor scenes, enabling generalization across a wide range of locations, including residential, office, shopping, leisure, and public spaces. Geolocation data, provided either as GPS coordinates or text-based location labels, is included alongside textual information such as descriptions and metadata as supplementary features. This multimodal approach enhances the dataset's versatility, particularly for tasks that benefit from both visual and textual data. It is important to note that the dataset does not explicitly identify specific locations in the manner typical of place recognition tasks. However, the accompanying text, and descriptions may contain useful information that could inform place recognition applications.

### 3.1 Data sources and collection methods

The INDOOR-3.6Mdataset was constructed using three primary sources: Flickr (flickr.com, 2024), a popular photo-sharing platform where users upload and tag images with metadata; Wikidata (wikidata.org, 2024), a free, collaborative knowledge base that provides structured data to support Wikipedia and other Wikimedia projects; and Booking.com (booking.com, 2024), a popular hotel booking website. For the image repositories (Flickr and Wikidata), we formulated search terms based on indoor scene categories from the Places365 dataset[1], and appended "indoor" to categories typically associated with outdoor environments. To ensure usability and proper attribution, we restricted our search to images with Creative Commons licenses and included only those with latitude and longitude coordinates. However, the initial search terms yielded few results because of these constraints. To address this, we manually refined the search terms, generalizing specific categories like "ski resort" to broader terms such as "resort" and expanding our vocabulary with synonyms and colloquial terms. Additionally, we introduced new categories that seemed relevant but were absent from the original list. Productive search terms included "living room", "indoor", "villa", "cottage", "diner", "office space", and "beach house". For the web scraping component, we employed country labels to initialize a crawler that retrieved images from search results for each country.

This collection process yielded approximately 10 million candidate images combined. In addition to visual data, we also collected associated textual metadata from these platforms, such as user-generated tags, descriptions, and captions. The textual data varies significantly in length, language, and detail, ranging from brief labels to detailed narratives or contextual information.

To ensure the dataset's focus remained on indoor scenes, we filtered the candidate images using the Places365 ResNet indoor/outdoor image classifier (Zhou et al., 2017). Recognizing that the distinction between indoor and outdoor scenes can sometimes be blurred, we used the classifier to quantify the *"indoorness"* of each image. We retained only those images with a probability of being indoor, $P(indoor) \geq 0.5$. Additionally, we recorded this likelihood score for each image in the metadata, placing images on a continuum between relatively indoor spaces ($P(indoor) = 0.5$) and purely indoor spaces ($P(indoor) = 1.0$).

### 3.2 Scale and Distribution

The INDOOR-3.6Mdataset comprises 3.6 million images spanning a wide variety of scenes from 223 countries worldwide, uploaded between 1978 and 2024. While the dataset aims to be representative of indoor environments, it is not entirely geographically representative due to its reliance on online sources (See Figure 2a). This dependence introduces inherent biases in geographic distribution, resulting in over-representation of regions with a strong digital footprint and larger populations such as United States (which represent 30% of the data), and under-representation of areas with less online activity or smaller populations. Figure 3a illustrates the dataset's distribution according to the MIT indoor scenes label set. A significant portion of the images are labeled as "tv studio", which predominantly corresponds to spaces where a TV is present—-commonly living rooms.

### 3.3 Metadata enrichment

The dataset incorporates metadata enrichment encompassing geospatial information, scene classification, and object segmentation. Using the GPS data, we use the Nominatim API(Nominatim, 2024) to perform reverse geocoding, yielding detailed location information including building names, street addresses, suburbs, and cities. This granular metadata facilitates fine-grained, location-based classification tasks. In addition, for each image, we include top 10 scene category labels obtained from Places365 and a ViT trained on MIT indoor Scenes dataset, as well as segmentation masks extracted using Segment Anything Model (SAM)(Kirillov et al., 2023) for pixel-level segmentation and YOLOv8(Jocher et al., 2022) for object detection and labeling. Scene labels, segmentation masks, and object detection results enhance the dataset by providing additional cues for geolocation. These annotations help models identify important features like furniture, signage, or cultural artifacts, which are critical for pinpointing locations. Such features also align with real-world practices, like Europol's 'Trace an Object'tra initiative, where visual clues in scenes are used to infer loca-

---

[1]`https://github.com/CSAILVision/places365/blob/master/categories_`
`places365.txt`

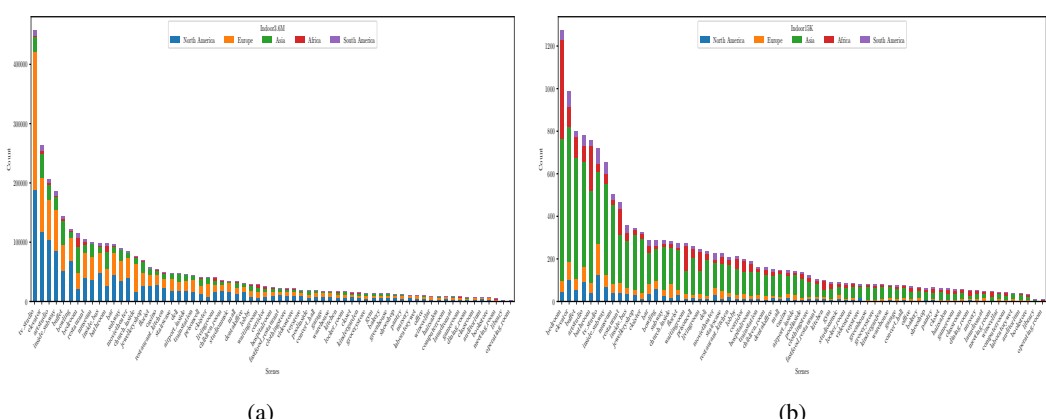

(a)            (b)

Figure 3: Side-by-side comparison of scene distribution in INDOOR-3.6M and INDOOR-15K.

tions. By including these annotations, the dataset supports more advanced and accurate geolocation methods.

# 4   INDOOR IMAGE GEOLOCATION BENCHMARK DATASET

Our analysis reveals that current benchmark datasets for image geolocation predominantly consist of outdoor scenery. Figure 4 illustrates the percentage of indoor images identified at various likelihood thresholds across existing mixed-environment image geolocation benchmark datasets. Furthermore, Table 2 demonstrates the performance variation of a pretrained GeoClip model (Vivanco Cepeda et al., 2024)—the current state-of-the-art for mixed environments—when applied separately to indoor and outdoor environments within these benchmark datasets. The results indicate that the model's performance degrades significantly when moving from outdoor to indoor settings. For instance, in the IM2GPS dataset, the average distance error for indoor images is 1,761.54 km compared to 1,079.67 km for outdoor images. This difference in error of approximately 700 km is substantial in the context of global positioning. To provide a tangible reference, this error is comparable to the east-west distance of Germany (approximately 640 km), illustrating the magnitude of the discrepancy between indoor and outdoor geolocation accuracy.

To address the limitations of current benchmark datasets, which predominantly focus on outdoor environments, we introduce a new benchmark dataset specifically for indoor geolocation: INDOOR-15K. This dataset is curated to minimize the visual biases of existing benchmarks by providing a diverse collection of 15,000 images from various indoor environments across 193 countries. To ensure the dataset is distinct from those used to train existing geolocation models, we carefully selected images captured after 2017—following the release of YFCC100M—and exclusively sourced

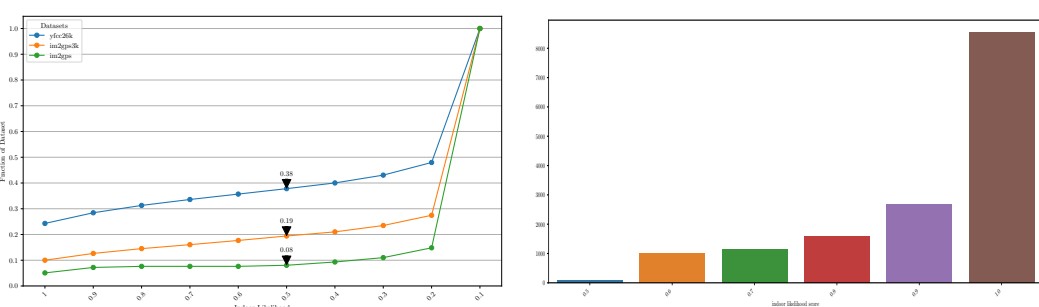

Figure 4: Cumulative distribution of indoor image probabilities

Figure 5: Distribution of images in INDOOR-15K by indoor likelihood scores.

Table 2: Accuracy and Average Distance Error for Indoor and Outdoor Images in current Geolocation Benchmark datasets, using GeoCLIP

| Dataset | Env. | Street (1km) | City (25km) | Region (200km) | Country (750 km) | Continent (2500 km) | Mean Dist. Error (km) |
|---------|------|--------------|-------------|----------------|------------------|---------------------|----------------------|
| IM2GPS | Indoor | 0.15 | 0.36 | 0.42 | 0.57 | 0.84 | 1761.53 |
|  | Outdoor | 0.17 | 0.42 | 0.62 | 0.78 | 0.90 | 1079.67 |
| IM2GPS3K | Indoor | 0.08 | 0.19 | 0.29 | 0.48 | 0.72 | 2618.79 |
|  | Outdoor | 0.14 | 0.35 | 0.52 | 0.70 | 0.84 | 1563.74 |
| YFCC26K | Indoor | 0.06 | 0.11 | 0.19 | 0.40 | 0.65 | 3179.86 |
|  | Outdoor | 0.11 | 0.24 | 0.41 | 0.64 | 0.81 | 1959.14 |

from booking.com, ensuring each image contains GPS metadata. This curation process resulted in a initial pool of approximately 800,000 images, from which we sampled the final benchmark set according to the methodology outlined in the next section.

## 4.1 SAMPLING STRATEGY

Our sampling methodology integrates both population density and land area to ensure a representative distribution of GPS points across countries. This approach accounts for the fact that countries with larger populations should receive proportionally more sampling points, while also considering the spatial diversity inherent in nations with expansive land areas. We account for population in our sampling strategy because it serves as a proxy for the density of human-made structures and indoor environments. Highly populated areas are more likely to contain a diverse range of indoor spaces, such as residential buildings, commercial centers, and public facilities. The allocation of GPS points for each country is determined using the following formula: $S_i = \max\left\{N_{\min}, \frac{\alpha \cdot P_i + \beta \cdot A_i}{\sum_{i=1}^{n}(\alpha \cdot P_i + \beta \cdot A_i)} \cdot N\right\}$ where: $S_i$ is the sample size for country $i$, $P_i$ represents the population of country $i$, $A_i$ denotes the land area of country $i$ in square kilometers, $\alpha$ is the weighting factor assigned to population, $\beta$ is the weighting factor assigned to land area, $N$ is the total number of GPS points to be sampled across all countries, $n$ is the total number of countries in the study, and $N_{min}$ is the minimum number of samples per country $i$, to prevent under-sampling

This formulation allows for the calibration of sample sizes based on the relative importance of population and land area through the parameters $\alpha$ and $\beta$. For example, setting $\alpha = 1$ and $\beta = 0$ results in a sampling strategy driven exclusively by population, while setting $\alpha = 0$ and $\beta = 1$ yields a distribution solely based on land area.

In constructing our test set, we chose $N_{min} = 3$, $\alpha = 0.3$ and $\beta = 0.7$, favoring land area over population in determining the sample size for each country. Population and land area data were obtained from publicly available sources provided by the World Bank (The World Bank, 2024). Once the number of points per country was determined, we sample uniformly within each country's available data points. The weighting factors of 0.7 for land area and 0.3 for population were chosen to prioritize geographic diversity, as represented by land area. As a result, our approach prevents the over representation of small, densely populated countries and the under representation of large, sparsely populated nations.

The choice of population and land area as proxies for scene visual diversity reflects the idea that highly populated countries tend to feature a broader range of indoor environments, shaped by diverse cultural, economic, and other social. Similarly, larger countries encompass varied geographic regions, often translating into more diverse architectural and interior styles. These provide a practical heuristic for achieving geographic balance without requiring additional data collection.

Our sampling strategy resulted in a dataset containing 15,025 GPS points, offering improved spatial representation of indoor imagery compared to existing benchmark datasets such as IM2GPS3K (3,000 points) and YFCC26k (26,000 points). Figure 6 illustrates the improved spatial distribution achieved through our methodology.

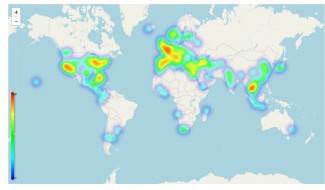 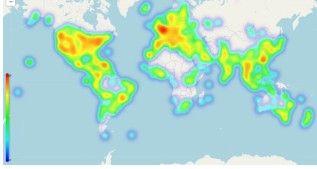 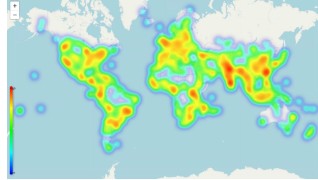

(a) Distribution of indoor images in IM2GPS3K

(b) Distribution of indoor images in YFCC26K

(c) Distribution of Indoor15K

Figure 6: Comparison of images with ($p_{indoor} \geq 0.5$) distributions across three datasets: Im2GPS3k, YFCC26k, and Indoor15k (ours).

## 4.2 EXPERIMENTS

In this study, we fine-tune GeoCLIP to establish a baseline for indoor geolocation using a subset of the INDOOR-3.6M dataset, following the sampling strategy described. GeoCLIP was selected for its state-of-the-art performance in environment-agnostic geolocation. We retained most of the training parameters from Vivanco Cepeda et al. (2024), including a constant learning rate of 1e-6 and a batch size of 256. The model converged after 10 epochs and outperformed the original GeoCLIP on our test set. Table 3 highlights the improved performance across all levels of granularity.

We also assessed the zero-shot classification performance of CLIP on a location classification task. For this, using the INDOOR-3.6M dataset, we divided the Earth into semantic geocells based on the approach in Haas et al., ensuring each geocell contained between 1,000 and 2,000 images. This resulted in approximately 1,300 geocells. We utilized the image encoder from the clip-vit-large-patch14 Radford et al. (2021) architecture to perform zero-shot classification of geocells. The encoder extracted visual embeddings, which were then used to predict geocells without additional training. For GPS prediction, the latitude and longitude of an image were approximated by averaging the GPS coordinates of all images within the predicted geocell. The results of these experiments are presented in Table 3.

The study underscores the potential of domain-specific training in enhancing geolocation models, particularly for indoor environments. Our experiments with GeoCLIP on the INDOOR-3.6M dataset reveal critical insights into model performance across various geographic scales, with the fine-tuned GeoCLIP consistently outperforming its counterparts. The reduction in mean distance error from 4089.11 km for the baseline GeoCLIP to 3598.02 km for the fine-tuned version is especially remarkable given the inherent complexity of indoor geolocation. The most striking observations emerge at broader scales, where fine-tuned GeoCLIP demonstrates pronounced gains, such as improving continent-level accuracy from 53% to 61% and country-level accuracy from 25% to 35%. These results highlight the ability of the model to leverage the diversity and richness of INDOOR-3.6M to capture geographically meaningful features. While the gains at finer scales, such as street and city levels, are more modest, the consistent improvements across all levels reinforce the importance of domain-specific datasets in overcoming the unique challenges of indoor geolocation.

To evaluate the impact of the proposed sampling strategy, we conducted ablation studies using datasets prepared with random sampling and the strategic sampling methodology described in the Appendix. The model finetuned on the dataset created using our sampling strategy yields better performance on geolocating both over represented classes and underrepresented classes.

Table 3: Comparison of GeoCLIP, fine-tuned GeoCLIP, and zero-shot CLIP on Indoor15K.

| Model | Street (1km) | City (25km) | Region (200km) | Country (750 km) | Continent (2500 km) | Mean Dist. Error (km) |
|---|---|---|---|---|---|---|
| GeoCLIP | 0.01 | 0.04 | 0.10 | 0.25 | 0.53 | 4089.11 |
| Zero-shot CLIP vision | 0.05 | 0.16 | 0.19 | 0.38 | 0.56 | 3812.86 |
| Finetuned GeoCLIP | 0.03 | 0.11 | 0.19 | 0.35 | 0.61 | 3598.02 |

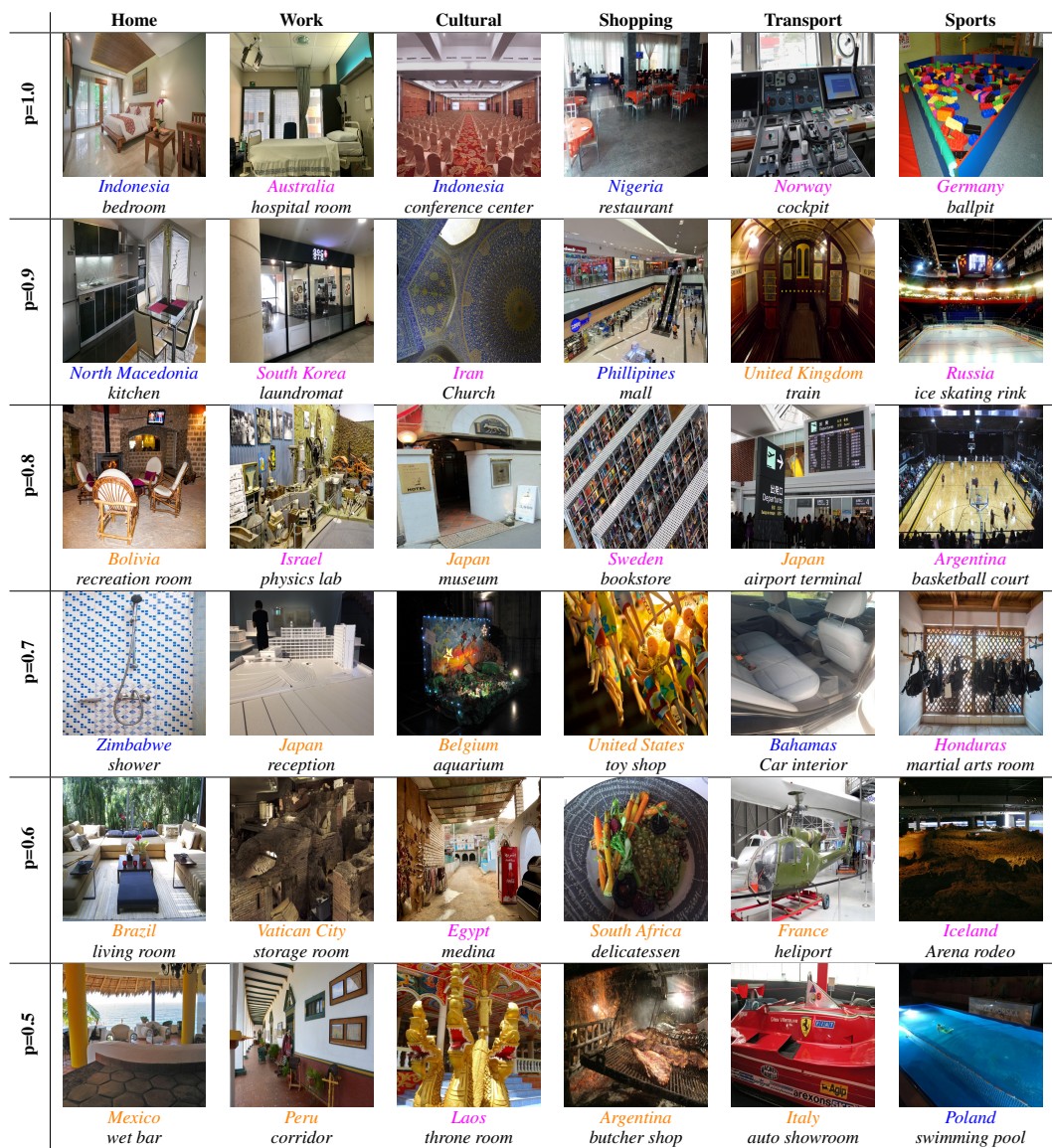

Figure 7: Samples of images from the dataset representing different parts of the world. The rows correspond to the indoor likelihood score $P(\text{indoor})$, while the columns categorize the scene types according to Places365 indoor scene categories at Level 2. Country names in blue, magenta, and yellow are sourced from Booking.com, Wikidata, and Flickr, respectively.

## 5 CHALLENGES

The development and utilization of large-scale indoor image datasets for geolocation present several challenges. Firstly, the INDOOR-3.6M dataset, like many large-scale datasets sourced from online platforms, exhibits significant **geographic and demographic biases**. This bias arises from the over representation of regions with higher internet penetration, tourism, and socioeconomic status, leading to under-representation of areas with limited digital footprints. This imbalance hinders the performance of geolocation models in underrepresented regions. Another critical issue is the **validation of GPS data**. In datasets sourced from user-uploaded images on photo-sharing sites, the accuracy of geotags could be unreliable. This can stem from a variety of factors, including device limitations, poor satellite coverage, or user errors in manually tagging locations. Since the GPS data in these platforms cannot be easily verified or cross-checked for accuracy, this remains a pervasive problem across geolocation datasets which rely on user-generated content, potentially leading to

discrepancies between model predictions and the true locations. Hotel booking and rental platforms provide more reliable and verifiable GPS information, but are limited to residential scenes.

The emergence of large vision models like Vision Transformers (ViTs) (Dosovitskiy et al., 2020) and CLIP (Radford et al., 2021) introduces challenges related to potential **data leakage**. These models are pretrained on vast datasets scraped from the internet, including Flickr-sourced collections like YFCC100M. Consequently, there is no guarantee that new publicly sourced datasets, such as INDOOR-3.6M, do not introduce a data leak when fine-tuning such models. This overlap could artificially inflate performance metrics during model evaluation. To mitigate this risk for our benchmark test set, we deliberately selected images captured after 2017, the publication year of YFCC100M, reducing the likelihood of overlap with this widely used dataset.

Indoor geolocation datasets introduce additional difficulties for geolocation systems due to **intra-class variation**. Unlike outdoor environments, where variations are often limited to views in the four cardinal directions (North, South, East, and West), indoor spaces exhibit far more complexity. In settings like hotels, different floors and rooms have distinct layouts, styles, and views, making it harder for models to establish consistent visual cues. This issue is exacerbated by the absence of clear landmarks, necessitating more nuanced feature extraction. Moreover, indoor environments are more subject to **temporal dynamics**. Frequent renovations, redecorations, and repurposing result in visual instability, which can quickly render models obsolete. Continuous updating or adaptive learning is required to ensure that models remain effective over time. To truly advance the field of indoor geolocation, it is crucial for future work to actively confront these issues, ensuring that models are both reliable and adaptable across diverse and evolving environments.

## 6 ETHICS

The INDOOR-3.6M dataset has been developed with careful attention to ethical considerations. The dataset contains geotagged indoor images sourced from public platforms, without the intention of identifying specific individuals or private spaces. We provides URLs and metadata information, rather than the raw image files, to prevent direct misuse, protect privacy, and avoid unauthorized redistribution of sensitive content. License and owner information from included to allow proper attribution. Geographic bias is acknowledged, particularly the over-representation of urban areas, and researchers are encouraged to apply sampling strategies and imbalance mitigation techniques to achieve fairer regional representation in model training. The dataset is *strictly* for research purposes, and misuse for purposes such as unauthorized surveillance or invasive applications is strongly discouraged. Researchers are urged to handle the data responsibly, especially during algorithm development and when implementing public-facing technologies.

There are concerns about the harmful applications of this dataset for geolocation technology, including privacy violations and unauthorized surveillance. We encourage researchers to remain mindful of the societal impact of their work, implementing safeguards to prevent abuse and adhering to privacy laws and ethical standards. It is essential that the research community stays actively engaged in discussions about the ethical development and use of indoor geolocation technologies, ensuring that advancements prioritize individual privacy and security. Misuse for invasive purposes is explicitly discouraged.

## 7 CONCLUSION

We introduce a new specialised dataset for indoor image geolocation (INDOOR-3.6M) as well as a benchmark dataset–INDOOR-15K. These contributions represent a significant step toward addressing the unique challenges of indoor image geolocation, where traditional outdoor models often struggle. Our dataset offers global coverage of diverse indoor spaces, enabling geolocation models to learn fine-grained features that are critical for accurately predicting the locations of indoor scenes. Our results demonstrate the utility of this dataset in improving the performance of geolocation models on indoor environments. Fine-tuning the GeoCLIP model with INDOOR-3.6M yielded measurable improvements across various levels of geographic granularity. However, indoor geolocation remains a challenging problem, with mean distance errors on the INDOOR-15K test set still exceeding 3,000 km. Despite these challenges, INDOOR-3.6M lays a strong foundation for advancing indoor geolocation.

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

# A APPENDIX

## A.1 SAMPLING STRATEGY

To evaluate the impact of the proposed strategic sampling methodology, we conducted experiments comparing its performance with that of random sampling. Both sampling methods were used to prepare three datasets for fine-tuning the GeoCLIP model. Performance was assessed across multiple geographic granularities, ranging from street-level (1 km) to continent-level (2500 km), and further analyzed for regions with high and low data representation to highlight the strengths and limitations of each approach. The averaged results are presented in Tables 4 and 5.

The results indicate that the proposed sampling method outperforms random sampling at finer granularities, such as street and city levels, in high-representation regions. For example, the proposed method achieves a street-level accuracy of 0.05 compared to 0.03 for random sampling and a city-level accuracy of 0.13 compared to 0.11. This improvement suggests that the proposed method's balanced geographic representation allows the model to capture features more effectively even in data-dense areas.

At coarser scales, such as country (750 km) and continent (2500 km), the differences between the two strategies become less pronounced. Both methods yield similar performance, with the proposed method achieving a slight edge in continent-level accuracy (0.70 vs. 0.71 for random sampling).

In low-representation regions, the proposed sampling method significantly improves performance at coarser granularities. For instance, at the region (200 km) level, the proposed method achieves an accuracy of 0.14 compared to 0.07 for random sampling, highlighting its ability to mitigate geographic biases and improve generalization to underrepresented areas. This trend continues at the country and continent levels, where the proposed method reduces errors by maintaining better spatial coverage.

Overall, the proposed method demonstrates consistent improvements at finer scales and excels in addressing biases in underrepresented regions, making it a valuable tool for creating datasets for geo-spatial applications.

Table 4: Performance Comparison of Random and Proposed Sampling Across Geographic Levels

| Sampling Strategy | Street (1km) | City (25km) | Region (200km) | Country (750 km) | Continent (2500 km) | Mean Dist. Error (km) |
|---|---|---|---|---|---|---|
| Random | 0.02 | 0.08 | 0.17 | 0.33 | 0.60 | **3577.67** |
| Proposed | **0.03** | **0.11** | **0.19** | **0.35** | **0.61** | 3598.02 |

Table 5: Performance on Overrepresented and Underrepresented Countries

| Sampling Strategy | Country representation | Street (1km) | City (25km) | Region (200km) | Country (750 km) | Continent (2500 km) | Mean Dist. Error (km) |
|---|---|---|---|---|---|---|---|
| Random | High | 0.03 | 0.11 | 0.21 | 0.39 | **0.71** | **2731.33** |
| Proposed | High | **0.05** | **0.13** | **0.22** | **0.42** | 0.70 | 2803.57 |
| Random | Low | 0.00 | 0.00 | 0.07 | 0.12 | 0.17 | 5019.10 |
| Proposed | Low | 0.00 | 0.00 | **0.14** | **0.15** | **0.28** | **4758.74** |

## A.2 EVALUATION OF FINETUNED GEOCLIP ON CURRENT GEOLOCATION BENCHMARK DATASETS

Table 6: Street to Continent-Level Accuracy and Average Distance Error for Indoor Images in current Geolocation Benchmark datasets, using GeoCLIP and finetuned GeoCLIP

| Dataset | Env. | Street (1km) | City (25km) | Region (200km) | Country (750 km) | Continent (2500 km) | Mean Dist. Error (km) |
|---|---|---|---|---|---|---|---|
| IM2GPS | GeoCLIP | **0.15** | **0.36** | 0.42 | 0.57 | 0.84 | 1761.53 |
| | Finetuned GeoCLIP | 0.11 | 0.31 | **0.47** | **0.78** | **0.94** | **910.37** |
| IM2GPS3K | GeoCLIP | 0.08 | 0.19 | 0.29 | 0.48 | 0.72 | 2618.79 |
| | Finetuned GeoCLIP | **0.09** | **0.28** | **0.46** | **0.65** | **0.82** | **1805.32** |
| YFCC26K | GeoCLIP | 0.06 | 0.11 | 0.19 | 0.40 | 0.65 | 3179.86 |
| | Finetuned GeoCLIP | **0.07** | **0.18** | **0.34** | **0.57** | **0.74** | **2360.12** |

Table 6 demonstrates that fine-tuning GeoCLIP improves its performance on indoor geolocation tasks, particularly at larger geographic scales (e.g., country and continent), with notable reductions in mean distance error across all datasets. However, improvements at finer scales, such as street-level accuracy, are limited, highlighting potential areas for further optimization.

