# OpenReview forum: "INDOOR-3.6M : A Multi-Modal Image Dataset for Indoor Geolocation"
_ICLR.cc/2025/Conference — ICLR 2025 Conference Withdrawn Submission_

### Official Review · Reviewer_jNt5 · 2024-10-21

**Soundness:** 2
**Presentation:** 1
**Contribution:** 3
**Rating:** 3
**Confidence:** 3

**Summary:**

This paper introduces a new dataset for indoor geolocation and proposes a sampling method to obtain a test set. Additionally, it demonstrates the fine-tuning of GeoCLIP on the proposed dataset.

**Strengths:**

1) Authors focus on a meaningful and niche (relative to large-scale outdoor scenarios) problem.
2) Building datasets is almost always beneficial, especially large-scale datasets.

**Weaknesses:**

1) The writing of this article needs to be improved, and the logic within each section is somewhat confusing. For example, the turn on L49 is not rigorous and only shows that you know little about the datasets for other tasks.
2) The article lacks citations for many key statements.
L41: such as seasonal changes, time of day, weather conditions, and human-induced modifications.
3) As I said in the “Strengths”, indoor geolocation (localization) is a meaningful research topic. However, the author lacks a rigorous literature review. I can name many famous indoor localization datasets without thinking: Baidu Mall(CVPR'17), InLoc(CVPR'18), NL-Indoor(CVPR'21). (Although the task objectives of these datasets are different from yours, I think they should be discussed.)
4) The experiments based on the proposed dataset are very limited and do not demonstrate the significance of the dataset.
5) Although you crawled the text descriptions of images, the word "multimodal" in the title is hardly seen again in the paper.

**Questions:**

None.

---

> ### Author Response · Authors · 2024-11-28
> **Response to Reviewer Comments**
>
> > The article lacks citations for many key statements. L41: such as seasonal changes, time of day, weather conditions, and human-induced modifications.
>
> The mentioned factors—seasonal changes, time of day, weather conditions, and human-induced modifications—stem from the authors’ experience and knowledge about the task. Additionally, these factors are mentioned in related work, such as Pramanick et al.[1], which explicitly mentions how seasonal variations and time of day can impact geolocation accuracy. We have included this citation in the revised manuscript to strengthen the statement.
>
> >As I said in the “Strengths”, indoor geolocation (localization) is a meaningful research topic. However, the author lacks a rigorous literature review. I can name many famous indoor localization datasets without thinking: Baidu Mall(CVPR'17), InLoc(CVPR'18), NL-Indoor(CVPR'21). (Although the task objectives of these datasets are different from yours, I think they should be discussed.)
>
> Thank you for highlighting the importance of a rigorous literature review and for mentioning datasets like Baidu Mall, InLoc, and NL-Indoor. While these datasets are indeed significant in their respective domains, they focus on specific tasks which differ from the scope of our work. Given these distinctions, we have chosen to focus our review on works and datasets directly relevant to global geolocation, i.e identifying where in the world an indoor image was captured, to ensure a clear and focused narrative in the manuscript. This approach allows us to better contextualize our contributions within the specific domain of geolocation research, rather than discussing datasets with fundamentally different objectives.
>
> > The experiments based on the proposed dataset are very limited and do not demonstrate the significance of the dataset.
> We have updated the experiments section with of the revised manuscript which we hope can be kindly considered for this review.
>
> > The experiments based on the proposed dataset are very limited and do not demonstrate the significance of the dataset.
>
> We selected GeoCLIP as the primary baseline due to its state-of-the-art performance in hybrid geolocation tasks, making it an ideal candidate to adapt to the unique challenges of indoor geolocation. By fine-tuning GeoCLIP on INDOOR-3.6M, we showcased how our dataset enhances the capabilities of a leading model, setting a benchmark for indoor-specific geolocation. Our focus on discussing visual data for geolocation aligns with real-world scenarios where images are often the sole resource, such as in forensic investigations.
>
> 1. Pramanick, Shraman, et al. "Where in the world is this image? transformer-based geo-localization in the wild." European Conference on Computer Vision. Cham: Springer Nature Switzerland, 2022.

---

> ### Comment · Reviewer_jNt5 · 2024-11-28
>
> Thank you for your reply!
>
> To be honest, Indoor-3.6M was the paper I was most eager to review during the bidding phase. However, I am somewhat disappointed overall (in terms of figures, presentation, experiments, and other aspects).
>
> I agree with your first two points, and this might indeed be just my personal opinion.
>
> However, regarding the third point, I maintain my view that adding experiments would enhance your contribution and help other researchers better understand this work. If you had provided additional experiments, I might have raised my score to 5 or 6. Unfortunately, you did not.
>
> I chose to lower my confidence level to reflect an increase to a score of 4.

---

> ### Comment · Reviewer_jNt5 · 2024-11-28
>
> Select more than three geolocalization methods (e.g. PIGEON) for experimentation, and refer to the supplementary materials of CLIP to explore the performance of this dataset on other tasks. This would greatly enhance the impact of this work.

---

### Official Review · Reviewer_fgUm · 2024-10-23

**Soundness:** 1
**Presentation:** 2
**Contribution:** 2
**Rating:** 3
**Confidence:** 3

**Summary:**

This paper introduces a novel image geolocation dataset tailored for indoor scenes, addressing the limitations of existing datasets and establishing a benchmark for evaluating indoor image geolocation algorithms. Specifically:
1. The paper presents a dataset that covers a wide variety of indoor scenes and includes rich multimodal metadata, which is expected to advance the field of indoor image geolocation;
2. The paper proposes an innovative sampling method to obtain geographically representative samples from datasets with geographic bias;
3. The paper provides a standardized evaluation framework for fair assessment and comparison of research progress in indoor geolocation research.

**Strengths:**

i) The dataset is collected from three different sources, covering a variety of indoor scenes, thus filling the gap in indoor image geolocation datasets and providing rich multimodal information;
ii) A sampling method that integrates population density and land area is proposed to ensure that the distribution of GPS points across regions is geographically representative;
iii) A new benchmark dataset specifically for indoor geolocation is introduced to address the limitation of existing benchmark datasets, which primarily focus on outdoor environments.

**Weaknesses:**

(1) The contribution of the dataset does not seem particularly prominent, especially regarding the limitations of existing datasets mentioned in the Introduction (such as insufficient diversity, imbalanced distribution, and blurred boundaries between indoor and outdoor environments), which have not been significantly addressed;
(2) Compared to some of the datasets in Table 1, the proposed dataset does not show a clear advantage in terms of scale.  Additionally,  I would like to know the amount of indoor scene data within mixed-scene datasets (e.g., YFCC100M). It seems feasible to separate indoor and outdoor images in such datasets using image classification methods.
(3) The experiments in Table 3 do not adequately demonstrate the superiority of the proposed dataset for this task. It is recommended to supplement the results by providing the performance of IndoorGeoCLIP on the three datasets listed in Table 2, to further substantiate the advantages of the proposed dataset.

**Questions:**

1.The paper demonstrates the performance of the IndoorGeoCLIP model on various levels of geolocation tasks (such as street-level, city-level, and country-level), but it lacks comparative experiments with other classic indoor geolocation methods, making it difficult to comprehensively validate the superiority of IndoorGeoCLIP.  Explicitly,
i) Lack of Baseline Model Comparisons: Apart from GeoCLIP, the experiments lack comparisons with other geolocation models, making it insufficient to illustrate the relative advantages of IndoorGeoCLIP in indoor scenes.
ii ) Insufficient Ablation Studies: The experiments only show the performance changes of the GeoCLIP model before and after fine-tuning, without conducting ablation analyses on the contributions of the dataset's multimodal features (such as textual and visual data) to geolocation.
2. Some of the images lack sufficient clarity.

---

> ### Author Response · Authors · 2024-11-28
> **Response to reviewer's comments**
>
> > The paper demonstrates the performance of the IndoorGeoCLIP model on various levels of geolocation tasks (such as street-level, city-level, and country-level), but it lacks comparative experiments with other classic indoor geolocation methods, making it difficult to comprehensively validate the superiority of IndoorGeoCLIP. Explicitly, i) Lack of Baseline Model Comparisons: Apart from GeoCLIP, the experiments lack comparisons with other geolocation models, making it insufficient to illustrate the relative advantages of IndoorGeoCLIP in indoor scenes. ii ) Insufficient Ablation Studies: The experiments only show the performance changes of the GeoCLIP model before and after fine-tuning, without conducting ablation analyses on the contributions of the dataset's multimodal features (such as textual and visual data) to geolocation.
>
> We focused on GeoCLIP as the primary baseline for comparison because it represents the state-of-the-art for hybrid geolocation tasks, encompassing both indoor and outdoor environments. GeoCLIP’s robustness and strong performance in mixed-environment geolocation make it a natural choice to establish a benchmark for our dataset. By fine-tuning GeoCLIP on INDOOR-3.6M, we were able to adapt a leading model to the unique challenges of indoor geolocation, demonstrating the effectiveness of our dataset while maintaining alignment with state-of-the-art methodologies.
> Our emphasis on the image as the primary input for geolocation tasks reflects real-world constraints where visual content is often the only resource available. In practical applications, such as forensic investigations, textual metadata is rarely accessible, making image-based models crucial.
> The inclusion of textual metadata alongside visual data expand the utility of INDOOR-3.6M beyond geolocation tasks. The rich textual annotations make the dataset a valuable resource for other computer vision challenges, such as scene understanding, multimodal representation learning, and image-to-text generation.
> We acknowledge that some images in INDOOR-3.6M may lack sufficient clarity, stemming from variations in resolution, lighting, and photographic quality. This was a deliberate choice to reflect the diverse types of imagery encountered in real-life geolocation applications, where input images often vary widely in quality.
>
> > Some of the images lack sufficient clarity.
>
> Including such images ensures that the dataset captures the realistic challenges faced by geolocation systems in practical use cases, such as forensic investigations, surveillance, and emergency response. These scenarios frequently involve suboptimal images, such as those taken with low-resolution devices or in poor lighting conditions. Training models on a dataset that includes such variability helps enhance their robustness and generalizability, enabling them to perform reliably regardless of the quality of the input image.

---

> > ### Comment · Reviewer_fgUm · 2024-11-28
> >
> > Thank you for your response. However, it seems that my concerns are not fully addressed, and the explanation is not entirely convincing. I also hope you can take into account some of the issues mentioned in the weaknesses.

---

### Official Review · Reviewer_G1Xc · 2024-10-26

**Soundness:** 2
**Presentation:** 3
**Contribution:** 2
**Rating:** 5
**Confidence:** 4

**Summary:**

This works introduce a new large-scale dataset to train and evaluate the tasks of global locations prediction from images. The train set consist of 3.6M image-text pairs collected from the internet on a global scale. The test set is constructed by sampling the images using the population and the land area of each contry. Finetuning the previous sota, GeoCLIP, on the proposed dataset achieves much better geolocation accuracy on the indoor scenes.

**Strengths:**

+ This paper propose a new geolocation dataset dedicating for indoor scenes. The evaluation results shows that the previous sota can perform even better on the indoor scenes after finetuning on the proposed dataset.
+ The discussions of many aspects of the geolocation tasks and the challenge of collecting the dataset is comprehensive.

**Weaknesses:**

The significance is somewhat unclear to me:
- Regarding dataset scale, it's unclear from Table1 that if the proposed dataset provide more indoor data than some of the other much larger scale mixed dataset like YFCC100M.
- The accuracy improvement for indoor geolocation task is limited to the proposed dataset. Can the fine-tuned models achieve better accuracy on the indoor subset of the other datasets?
- What is the accuracy merit by the proposed dataset additional to the existing resource? Say if we train on a combined indoor data from existing datasets (e.g., Im2GPS, YFCC100M, MP-16, Hotels50k), what is the additional accuracy boost by adding the proposed INDOOR-3.6M?

The sampling strategy in Sec.4.1 looks ad-hoc to me:
- The sampling weight of each contry is determine by the weighted sum of it's population and land area. How the final weight (population * 0.3 + area * 0.7) determined? As it listed as one of the main contribution, I expect more insights. Perhaps an analysis of the accuracy on different contries by varying the weighting can show some support for the goal of the sampling.
- The sampling weight seems to assume that the scene visual diversity of the contries are linearly propotion to the population and land area. Is there any support for this assumption? Perhaps some measure using CLIP distance can provide some insight.

**Questions:**

In Table1, what is the column "Benchmark" actually stand for? Is it saying that if the dataset has a train/test split?

Why the correctness of the city/contry/continent prediction is determined by a distance threshold (Table 2 and 3)?

Scene labels, segmentations, and object detection results are provided by the dataset. What is the purpose of them? Are they serve as some additional conditions for the geolocation task?

---

> ### Author Response · Authors · 2024-11-28
> **Response to Concerns on Dataset Scale, Accuracy Improvement, and Sampling Strategy**
>
> > it's unclear from Table1 that if the proposed dataset provide more indoor data than some of the other much larger scale mixed dataset like YFCC100M.
>
> Thank you for raising this point. While YFCC100M is significantly larger in overall scale, it was curated in 2016 and may not reflect recent changes in indoor environments. INDOOR-3.6M, on the other hand, includes images uploaded as recently as 2024, ensuring the dataset captures contemporary indoor spaces. This temporal relevance, combined with its exclusive focus on indoor imagery, provides a unique advantage over older, mixed-environment datasets like YFCC100M, ensuring relevance for real-world applications, such as human trafficking investigations, where up-to-date imagery is critical..
>
> > Can the fine-tuned models achieve better accuracy on the indoor subset of the other datasets?
>
> We show a table showing the peformance of the finetuned models on the other datasets in Appendix A.2
>
> > The sampling weight of each country is determined by the weighted sum of it's population and land area. How the final weight (population * 0.3 + area * 0.7) determined? The sampling weight seems to assume that the scene visual diversity of the contries are linearly propotion to the population and land area. Is there any support for this assumption? Perhaps some measure using CLIP distance can provide some insight.
>
> The use of population and land area as proxies for scene visual diversity is based on the idea that:
> - Population: Countries with larger populations are likely to have more diverse human-made structures and indoor environments, reflecting varied cultural, economic, and functional needs.
> - Land Area: Larger countries typically encompass more diverse geographic regions, which can translate into a wider variety of architectural and interior styles.
> While these factors are not perfect predictors of scene visual diversity, they offer a practical and scalable heuristic for balancing the dataset across countries. We acknowledge that this assumption could benefit from further validation. We will clarify this rationale and acknowledge the limitations of our current approach in the manuscript.
>
> It is worth noting that our sampling strategy is the first to address geographic bias without requiring additional data collection, making it a practical and resource-efficient solution. By leveraging existing data, we propose an accessible approach to mitigating over- or under-representation of certain regions. Future work will explore advanced methods, perhaps including visual diversity metrics such as CLIP feature distances, to further optimize sampling and refine our methodology.
>
>
> > In Table1, what is the column "Benchmark" actually stand for?
>
> The "Benchmark" column in Table 1 indicates whether the dataset provides a dedicated test or evaluation set specifically designed to benchmark the performance of geolocation models. While this often implies a clear train/test split, it may also include datasets that are predefined test sets for standardized evaluation without an explicitly defined training data.  We have clarified this definition in the revised manuscript to avoid any confusion.
>
> > Why the correctness of the city/contry/continent prediction is determined by a distance threshold
>
> Our choice of distance thresholds for evaluating geolocation predictions follows established practices in prior work on image geolocation, allowing direct comparison with state-of-the-art methods. Using distance thresholds as a metric ensures standardized evaluation and facilitates meaningful benchmarking across datasets and models. In addition, distance thresholds offer a gradient of accuracy that reflects the model's true geographical precision during evaluation, avoiding unfair assessments of models whose predictions are geographically close to the true location but may fall outside strict administrative boundaries.
>
>
> > Scene labels, segmentations, and object detection results are provided by the dataset. What is the purpose of them? Are they serve as some additional conditions for the geolocation task?
>
> These annotations enhance the dataset’s versatility and support further research in geolocation and related areas. They provide supplementary cues that can improve geolocation accuracy, such as segmentation masks to exclude irrelevant regions and scene labels or detected objects to identify localized furniture, signage, or cultural artifacts. Real-world geolocation experts, such as those in Europol’s “Trace an Object” initiative, frequently rely on objects in scenes to infer locations ​(See https://www.europol.europa.eu/media-press/newsroom/news/new-trace-object-uploads-fresh-leads-needed-in-child-sexual-abuse-cold-cases and https://www.europol.europa.eu/stopchildabuse). By including these features, the dataset enables researchers to explore models that mimic such expert strategies.

---

> > ### Author Response · Authors · 2024-11-28
> >
> > > I acknowledge the authors’ introduction of an interesting sampling strategy, but they have not demonstrated its advantages through appropriate ablation studies.
> >
> > We have included our ablation study to evaluate the sampling strategy’s effectiveness, as described in Appendix A.1 (lines 702–751) of the revised manuscript. These highlight our methodology's advantages in improving geographic representation and performance.
> >
> > > In the supplementary material, the download_images.py code essentially functions as an automated data-scraping script that downloads images through URLs. It raises the question of whether the authors have obtained proper authorization from all websites involved in the dataset to conduct automated data scraping.
> >
> > The download_images.py script serves as a tool for downloading *publicly* accessible images via URLs provided in the dataset metadata. As stated in our ethics section, this dataset is intended solely for research purposes and downloading these images should . We strongly discourage its use in any way or for any applications that may breach privacy, violate ethical standards, or contravene legal norms.

---

> > ### Comment · Reviewer_G1Xc · 2024-11-30
> >
> > Appreciate authors effort in the rebuttal.
> >
> > I'm still not fully convinced that the number of people and the land area are good indicator of the diversity of a country. It's possible that a smaller country have more different kind of people and culture comparing to a larger country due to some historical reason.
> >
> > I understand that previous work may use distance threshold for evaluation. It is still good to propose another metrics by using the actual continent/country/region to judge if the prediction is correct.
> >
> > By also checking the other review comments, I'm still lean to negative rating.

---

### Official Review · Reviewer_d91o · 2024-11-05

**Soundness:** 3
**Presentation:** 3
**Contribution:** 2
**Rating:** 3
**Confidence:** 4

**Summary:**

The paper introduces INDOOR-3.6M, a large dataset specifically for indoor image geolocation, addressing the challenges posed by indoor environments that lack the rich landmarks of outdoor spaces. INDOOR-3.6M includes 3.6 million globally diverse, geotagged indoor images, accompanied by metadata for enhanced model training. Alongside this, the authors provide INDOOR-15K, a benchmark dataset for evaluating indoor-specific geolocation models, and propose a sampling strategy to ensure balanced geographic diversity. They also propose a sampling method, which combines population density and land area to ensure balanced geographic representation within the INDOOR-15K benchmark dataset.

**Strengths:**

- The presentation is well-structured, making the paper clear and easy to understand.
- Unlike other datasets, the proposed dataset is easily accessible and distributable, as demonstrated in the supplementary materials.
- The proposed dataset fills a critical gap in the research community, providing much-needed resources for indoor geolocation tasks.

**Weaknesses:**

- In line 231, the paper mentions collecting images from the internet that contain latitude and longitude coordinates. Is there a human review mechanism to ensure the accuracy and reliability of the geographic information in these images? Additionally, even if the images themselves are under a CC license, is there a protocol to blur potential privacy-sensitive information within the images, such as faces, intimate clothing, etc.?
- Could the release of this dataset lead to illegal applications, such as using images (e.g., from social media) to obtain the user locations (even if they don’t want others to know their location), thereby introducing security risks?
- In line 490, the proposed dataset only provides URLs. These URLs may become inaccessible over time, especially for sites where links frequently change, such as booking websites (with some hotels even removing pages). How does the dataset plan to address the potential issue of broken or inaccessible URLs?
- Is it reasonable to determine geographic information solely from images? Have experiments been conducted for more complex cases? For example, (1) many hotel chains use standardized decor, so to what extent can images alone reliably confirm the location of, say, the same hotel brand in the U.S. versus China? (2) Different individuals may have distinct decor and layout styles that may be more indicative of personal taste than geographic location. For instance, a Chinese staff working in the U.S. might have a TV studio with a Chinese interior style. To what extent can models accurately recognize the geographic location in such cases?
- The paper lacks a detailed description and explanation of the proposed IndoorGeoCLIP model. Are the authors only fine-tuning the existing GeoCLIP model using their proposed dataset? If so, this contribution might be seen as insufficiently significant and could be considered merely a necessary experiment to support the dataset.
- The experiments seem insufficient. The authors should consider using more existing methods to evaluate the proposed Indoor15K benchmark; currently, Table 3 includes only GeoCLIP, which seems inadequate. Additionally, in Table 3, wouldn’t it be more intuitive to label IndoorGeoCLIP directly as “GeoCLIP (fine-tuning)”?
- I acknowledge the authors’ introduction of an interesting sampling strategy, but they have not demonstrated its advantages through appropriate ablation studies. I also believe this sampling method could be beneficial for other geolocation datasets (both indoor and outdoor), and further verification of its performance and feasibility would be valuable.
- In the supplementary material, the download_images.py code essentially functions as an automated data-scraping script that downloads images through URLs. It raises the question of whether the authors have obtained proper authorization from all websites involved in the dataset to conduct automated data scraping.

**Questions:**

Refer to Weaknesses.

---

> ### Author Response · Authors · 2024-11-28
> **Response to Concerns Regarding Data Collection, Dataset Reliability, Model Contributions, and Ethical Implications**
>
> > Is there a human review mechanism to ensure the accuracy and reliability of the geographic information in these images? Additionally, is there a protocol to blur potential privacy-sensitive information within the images, such as faces, intimate clothing, etc.?
>
> We appreciate the reviewer’s concern about the accuracy of geotags and the potential privacy implications of the dataset. We address these concerns:
> 1. Reliability of Geotags: The GPS tags of images sourced from Booking.com and Wikidata are highly reliable, as they are typically reviewed and verified by humans before being published. While the geotags from Flickr may be noisier, their inclusion is intentional to reflect real-world data conditions. Such variability provides an opportunity for researchers to develop robust models that can handle noisy geotags effectively, which is critical for real-world applications.
> 2. Privacy and Accessibility: The images included in the dataset are already publicly accessible and hosted on their original platforms, ensuring that we do not distribute sensitive or restricted content. Since these images are “in the wild,” they are already accessible to anyone under their respective licensing terms. Our approach merely aggregates links to these images, respecting their original hosting context.
> 3. Omission of Privacy-Sensitive Content: We provide segmentation mask labels for each image. These labels enable researchers to programmatically identify and exclude images containing people or other sensitive elements if required for their specific use case. This will allow researchers to customize the dataset to align with the ethical guidelines and privacy considerations for their specific application.
>
> > Could the release of this dataset lead to illegal applications, thereby introducing security risks?
>
>  We understand your concern about potential misuse and have explicitly outlined in the manuscript that the dataset is intended for research purposes and we strongly discourage any applications that could lead to privacy violations or unethical outcomes.
>
> >  How does the dataset plan to address the potential issue of broken or inaccessible URLs?
>
> While we recognize that some URLs may become unavailable over time, we anticipate this affecting only a small fraction of the dataset given its scale and diversity. Additionally, the accompanying metadata and contextual information remain valuable resources for research, even if some images are no longer accessible. To ensure data integrity, we will include the MD5 checksum of each image to the dataset to ensure that downloaded images are identical to the ones in our proposed dataset.
>
> > Is it reasonable to determine geographic information solely from images?
>
> We appreciate the reviewer’s observations on the challenges of inferring geographic information from images. While humans may struggle to pinpoint exact locations, they excel at estimating approximate regions using semantic reasoning and data association [1,2], highlighting the feasibility of geolocation as a computational task.
> Crowdsourced geolocation campaigns (see https://x.com/Europol/status/895978347263668224 and https://www.reddit.com/r/TraceAnObject/comments/lel9f9/tao17390_07feb2021_can_you_identify_this_hotel/), demonstrate how subtle visual markers enable humans to locate scenes, providing a basis for training models to replicate and enhance these capabilities. Prior work, like Stylianou et al.’s Hotels50K, shows that even in standardized environments, localized visual cues support accurate geolocation. Our INDOOR-3.6M dataset builds on this by offering geographically diverse images with features like objects, text, and scene-specific details that aid geolocation by capturing regionally distinctive cues.
> While standardized decor or culturally specific styles pose challenges for models, INDOOR-3.6M is specifically designed to address these complexities, enabling models to learn nuanced visual cues and improve geolocation accuracy, even in ambiguous scenarios.
>
> >  The paper lacks a detailed description and explanation of the proposed IndoorGeoCLIP model.
> >  wouldn’t it be more intuitive to label IndoorGeoCLIP directly as “GeoCLIP (fine-tuning)”?
>
> Your suggestion to label IndoorGeoCLIP as “GeoCLIP (fine-tuning)” for clarity is valid and we have implemented this in our revised manuscript. Regarding the inclusion of additional geolocation methods for benchmarking, we primarily focused on GeoCLIP as it is the current state-of-the-art for environment agnostic geolocation.
>
>
> 1. Hays, James, and Alexei A. Efros. "Large-scale image geolocalization." Multimodal location estimation of videos and images (2015): 41-62.
> 2. Kohler, Rachel. Supporting Open Source Investigative Journalism with Crowdsourced Image Geolocation. Diss. Virginia Tech, 2017.

---

> ### Comment · Reviewer_d91o · 2024-11-28
>
> Thanks for your rebuttal. I agree with your point that broken links may constitute a relatively small fraction in the future, but this largely depends on how long the “future” is defined. Furthermore, even the absence of a single image in the training set can potentially affect the reproducibility of every model’s results. Missing images in the test/validation set are equally critical, as they fundamentally alter the evaluation of models.
>
> I do not think that providing MD5 checksums is an effective solution to address this ‘data missing’ issue. From the perspective of datasets, the potential unavailability of data is a critical weakness that can severely limit their application in future research.

---

### Official Review · Reviewer_gNbp · 2024-11-05

**Soundness:** 3
**Presentation:** 3
**Contribution:** 3
**Rating:** 6
**Confidence:** 3

**Summary:**

This paper introduces INDOOR-3.6M, a large-scale dataset of geotagged indoor imagery for indoor geolocation tasks. Recognizing the limitations of existing geolocation models in indoor environments, the authors propose a new sampling methodology to ensure geographic diversity and balance in their dataset. They also introduce INDOOR-15K, a benchmark dataset specifically designed for evaluating indoor geolocation models. Lastly, they showcase the dataset's utility by introducing IndoorGeoCLIP, a fine-tuned version of the GeoCLIP model, which demonstrates superior performance compared to the baseline GeoCLIP on their test set.

**Strengths:**

- This paper addresses a relevant gap in image geolocation by creating a dataset specifically for indoor environments. Existing datasets primarily focus on outdoor scenes, making them unsuitable for the unique challenges of indoor geolocation. The creation of the dataset, along with the development of a specialized indoor geolocation benchmark (INDOOR-15K) are novel contributions.

- The dataset seems carefully curated and includes metadata, such as scene classification and object segmentation, which add to its potential applications. The use of the Places365 ResNet indoor/outdoor image classifier to filter images is good. Further, the effort to mitigate data leakage concerns by selecting images captured after 2017 for the INDOOR-15K benchmark demonstrates a commitment to quality and reliable evaluation.

- The paper is well-written and structured logically, making it easy to understand. The authors clearly articulate the challenges of indoor geolocation and provide a good overview of existing datasets and their limitations.

- The INDOOR-3.6M dataset has the potential for numerous interesting applications. The most direct is the development and evaluation of more robust and accurate indoor geolocation models. The dataset can also facilitate research in related areas, such as indoor navigation, scene understanding, and place recognition.

**Weaknesses:**

- While the paper introduces IndoorGeoCLIP as a specialized model fine-tuned on their dataset, the evaluation is limited. Exploring and comparing the performance of other state-of-the-art geolocation models or techniques on INDOOR-15K would strengthen the analysis. Additionally, the authors could include a more in-depth error analysis to identify the specific challenges posed by indoor geolocation, and what it can be used for, to guide future research.

- The paper primarily focuses on creating and describing the dataset. It lacks a thorough demonstration of the dataset's usefulness beyond the fine-tuning of GeoCLIP. Further experiments and analyses showcasing the dataset's application in tasks like place recognition, or indoor navigation would strengthen the paper significantly.

- The reliance on URLs to online sources for data access could lead to unreliable data availability, where links might become broken over time, limiting the dataset's long-term usability. The authors should consider providing alternative access methods, such as potentially mirroring the links.

**Questions:**

N/A

---

> ### Author Response · Authors · 2024-11-28
> **Response to suggestions for Enhancing Dataset Utility and Accessibility**
>
> > While the paper introduces IndoorGeoCLIP as a specialized model fine-tuned on their dataset, the evaluation is limited.
>
> We acknowledge the importance of a comprehensive evaluation and comparison with other geolocation models. We primarily focused on comparing with GeoCLIP to establish a baseline using our proposed dataset as GeoCLIP is the current state-of-the-art for environment agnostic geolocation.
>
> > The paper primarily focuses on creating and describing the dataset. It lacks a thorough demonstration of the dataset's usefulness beyond the fine-tuning of GeoCLIP. Further experiments and analyses showcasing the dataset's application in tasks like place recognition, or indoor navigation would strengthen the paper significantly.
>
> We also appreciate your suggestion to expand on the dataset's potential applications in tasks like place recognition etc. While it is true that datasets specialized for tasks like place recognition or indoor navigation exist, our primary focus is on indoor geolocation—a task for which no large-scale, geographically diverse dataset currently exists. INDOOR-3.6M was specifically designed to address this critical gap by providing the necessary scale, geographic diversity, and multimodal metadata to advance indoor geolocation research.
> Tasks like place recognition and navigation typically rely on datasets optimized for those purposes, but such datasets are often geographically or contextually limited. While our dataset includes multimodal metadata that can support broader applications, we believe that the most significant and immediate contribution of INDOOR-3.6M lies in the indoor geolocation domain, a domain where existing datasets fall short.
>
> > The reliance on URLs to online sources for data access could lead to unreliable data availability, where links might become broken over time, limiting the dataset's long-term usability. The authors should consider providing alternative access methods, such as potentially mirroring the links.
>
> Finally, we note your concern regarding URL-based data access and carefully chose this approach to balance data availability, ethical considerations, and potential misuse risks. By linking to the original hosts of the images, we respect copyright and data protection regulations while ensuring compliance with privacy laws. This approach aligns with datasets such as LAION-5B dataset[1] (https://laion.ai/faq/), OpenImages Dataset (https://storage.googleapis.com/openimages/web/download_v7.html#df-image-information) [2] which also provide URL-based linking, and face similar challenges.
> Importantly, hosting images solely on their original platforms introduces an additional safeguard against misuse. These platforms have established infrastructures to monitor and regulate access to their content, providing a layer of deterrence to bad actors. This is particularly critical given the potential harmful applications of geotagged indoor imagery. By ensuring that access to the images occurs through trusted hosts, we reduce the risk of the dataset being exploited for unethical purposes.
> While we recognize that some URLs may become unavailable over time, we anticipate this affecting only a small fraction of the dataset given its scale and diversity. Additionally, the accompanying metadata and contextual information remain valuable resources for research, even if some images are no longer accessible. To ensure data integrity, we will include the MD5 checksum of each image to the dataset to ensure that the downloaded images are identical to the ones in our proposed dataset.
>
> 1. Schuhmann, Christoph, et al. "Laion-5b: An open large-scale dataset for training next generation image-text models." Advances in Neural Information Processing Systems 35 (2022): 25278-25294.
> 2. Kuznetsova, Alina, et al. "The open images dataset v4: Unified image classification, object detection, and visual relationship detection at scale." International journal of computer vision 128.7 (2020): 1956-1981.

---

### Note · Authors · 2025-01-09

**Comment:**

Dear Reviewers,

Thank you for taking the time to review our manuscript and providing valuable feedback. Your insights have been immensely helpful in identifying areas where we can further enhance the clarity, impact, and utility of our work.

After careful consideration, we have decided to withdraw the paper to address your comments more comprehensively. We are committed to refining our contributions to better demonstrate our dataset's utility and strengthen the overall impact of the research.

Thank you once again for your time and thoughtful input.

**Withdrawal Confirmation:**

I have read and agree with the venue's withdrawal policy on behalf of myself and my co-authors.